# Comparison of the Vegetation Index of Reclamation Mining Areas Calculated by Multi-Source Remote Sensing Data

**Jiameng Hu [1], Baoying Ye [1,2], Zhongke Bai [1,2,3,*] and Jiawei Hui [1]**

[1]  School of Land Science and Technology, China University of Geosciences, Beijing 100083, China; 2112210002@email.cugb.edu.cn (J.H.); yebaoying@cugb.edu.cn (B.Y.); 2112190035@cugb.edu.cn (J.H.)
[2]  Key Laboratory of Land Consolidation and Rehabilitation, The Ministry of Natural Resources, Beijing 100035, China
[3]  Technology Innovation Center of Ecological Restoration Engineering in Mining Area, The Ministry of Natural Resources, Beijing 100083, China
[*]  Correspondence: baizk@cugb.edu.cn

**Abstract:** Following vegetation reclamation in mining areas, secondary damage may occur at any time, especially in locations that have been mined for decades or even hundreds of years. Effective monitoring strategies are required to accurately assess plant growth and to detect the ecological effects of reclamation. Single satellite monitoring is often difficult to ensure vegetation monitoring needs, therefore multi-source remote sensing is preferred. Different sensor parameters and variation in spectral bands can lead to differences in the type of data obtained, and subsequently, methods for evaluating these differences are required for simultaneous sensor/band use. In this study, NDVI was selected to characterize the vegetation growth of the Antaibao Open-pit Coal Mine Dump by analyzing the correlation between different types of sensors (Landsat 8, HJ, Sentinel-2) and vegetation greenness in order to facilitate satellites' replacement and supplement. Results show that: (1) Landsat 8 and Sentinel-2 satellite have a high relevance for monitoring the vegetation, but the correlation between these two sensors and HJ is relatively low, (2) the correlation between NDVI values varied by vegetation type, tree ($R = 0.8698$) > combined grass, shrub and tree ($R = 0.7788$) > grass ($R = 0.7619$) > shrub ($R = 0.7282$), and (3) the phenomenon of "Low value is high, high value is low" in the NDVI value with HJ satellite monitoring may have been caused by a weak signal strength and low sensitivity of the HJ sensor. Comparing the correlation of multi-source sensors to monitor the vegetation in the mining areas can be helpful to determine the alternative supplement of sensors through conversion formulas, which are helpful in realizing the long-term monitoring of dumps and detecting reclamation response in mining areas.

**Keywords:** multi-source sensors; NDVI; reclamation mining area; linear regression analysis



## 1. Introduction

Coal has played a vital role in meeting China's energy production demands. Coal has been critical for providing energy required for sustaining life and ensuring the production of good and services. In 2019, China industrial enterprises produced 3.75 billion tons of raw coal, an increase of 4.2% over the previous year [1]. While mineral resources provide the energy required for economic development, these also result in unprecedented severe environmental degradation, including air pollution, large-scale land disturbance, and damage to ecosystem resilience and sustainability [2,3]. To compensate for the ecological damage caused by mining activities, it is often accompanied by reclamation in the long-term mining process [4]. One of the most important steps in reclamation is vegetation restoration. By using effective monitoring techniques within mining areas, vegetation conditions can be reclaimed by enhancing restoration planning and implementation [5,6]. Relying only on traditional ground sampling experiments to monitor vegetation condition requires tremendous manpower and financial resources, which is not realistic. Compared

with traditional methods, remote sensing monitoring can accomplish large-scale vegetation monitoring and provide effective and timely vegetation information [6–13].

Although remote sensing monitoring is convenient, it is often difficult for single satellite data to ensure the continuity of time and space. Therefore, exploring the association between different satellites is conducive to supplementing the different types of satellite data, thus realizing long-term and all-around monitoring needs. There are many studies on the use of remote sensing technology to monitor vegetation growth to determine the correlation between different sensors, both locally and globally. Based on the measured reflectance data on the ground and the PROSALL model, Yang Fei et al. compared the LAI and fresh biomass of Environmental satellite and Landsat satellite data, and analyzed the influence of the main influencing factors between the two sensors [14]. Xu Guangzhi identified grasslands as the source area of the Beijing–Tianjin sandstorm as the research object. The author selected China satellite sensors, BJ-1 and HJ, and the American Landsat satellite data as the data sources, combined with the simultaneous ground measurement of grassland vegetation coverage, leaf area index and above-ground biomass data. The study systematically compared the differences in the three sensors in estimating the physiological parameters of grassland [15]. Based on multi-source remote sensing data, Kristin B. Byrd conducted a long-term historical time series along the Pacific Coast along with the plant community layer to analyze the impact of its changes on wetlands [16]. Due to the high cloud cover in the tropics, this limits the acquisition of optical remote sensing data. Amoakoh Alex O combined optical images with radar and elevation data and found that the integrated Sentinel-2, Sentinel-1, and SRTM datasets have the highest overall accuracy (94%) [17].

Judging from current research results, although there have been a large number of studies on the interactive comparison of different satellites vegetation monitoring, there are few comparative studies on monitoring the vegetation of dumps in mining areas. There are many differences between the vegetation within mining areas and ordinary vegetation. Coal mining can create large-scale ecological disturbances within mining areas, causing problems such as ground subsidence and vegetation degradation, which makes the vegetation ecosystem in the mining area highly unstable in the early stage of formation. Additionally, in the process of forming dumps, open-pit coal mining occupies a large amount of land, destroys the original landscape, affects the quality of local habitats, and poses a serious threat to the ecosystem. Therefore, the restoration and reconstruction of dumps have become the key factor in reclamation. It is very important to judge the restoration of the ecological environment by monitoring vegetation restoration. At present, there are more than 1500 open-pit mines in China, and large quantities of abandoned rocks and soils have been produced during the mining process. It is difficult to realize long-term monitoring of soil dumps through a single sensor, and multiple sensors are usually required to complement each other. In summary, comparing the correlation of multi-source sensor monitoring with vegetation in mining areas will help to accomplish long-term monitoring of dumps and to implement the reconstruction of ecological structure [18].

In this paper, NDVI was calculated by multi-source remote sensing satellites (Landsat 8, HJ, Sentinel-2) in order to monitor the vegetation growth of the Antaibao Open-pit Coal Mine in Shuozhou city, Shanxi Province. The regression of NDVI was then analyzed to reflect the correlation between different sensors and vegetation cover. In this way, the correlation coefficients and transformation equations of different sensors for monitoring the reclaimed vegetation were estimated to facilitate the replacement and supplement of different satellite sensor data in monitoring and reclamation assessment. This may play a reference role for multi-source and dynamic monitoring of the vegetation change within mining areas, which is beneficial for improving the accuracy and continuity of vegetation monitoring.

## 2. Materials and Methods

### 2.1. Study Area

The Antaibao Open-pit Coal Mine is located in Shuozhou city, Shanxi Province, 112°10′58″~113°30′ E, 39°23′~39°37′ N [19], in the east of the Loess Plateau. The climate is characterized as a continental monsoon climate. The landform type is a loess hill with an elevation that ranges from approximately 1300 to 1400 m. Seasons are distinct, characterized by lower rainfall and snow in the spring, concentrated precipitation in the summer, less rain in the autumn, and lower snowfall in the winter. The annual average temperature ranges between 5.4 and 13.8 °C, and average annual precipitation ranges between 428 and 449 mm.

The vegetation cover types of the four dumps are distinct. The vegetation coverage within the South Dump (SD) is dominated by tree species. The vegetation here has formed as tall tree bodies in a long-term stable environment, with roots that have an independent trunk and the trunk is distinct from the canopy. According to the actual site investigation, the main tree species is black locust (*Robinia pseudoacacia Linn.*), and also contains a small amount of sea buckthorn (*Hippophae rhamnoides Linn.*), caragana (*Caragana Korshinskii Kom*), and poplar (*Populus* L.). The West Dump (WD) is dominated by shrubs and contains a high density of sea buckthorn. Types of grasses include alfalfa (*Medicago sativa* L.) and erect milkvetch (*Astragalus adsurgens pall*) within the West Expansion Dump (WED). The Inner Dump (ID) is dominated by a mix of grasses, shrubs, and trees, covered by sea buckthorn, narrow-leaved oleaster (*Elaeagnus angustifolia Linn.*), elm (*Ulmus pumila* L.) and caragana. The differences between the four dumps are mainly reflected in the differences between vegetation cover. At the same time, invasions of different species may exist within a small scope. Based on this, the following four study areas were determined (Figure 1).

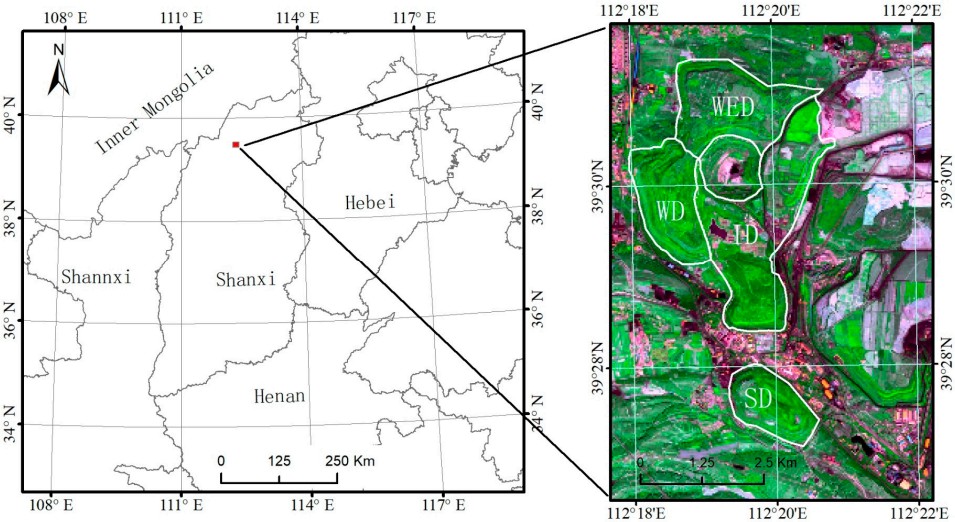

**Figure 1.** Location of the study area (Shuozhou City, Shanxi Province, 112°10′58″~113°30′ E, 39°23′~39°37′ N).

Currently, Antaibao is one of the largest open-pit coal mines in China. Its mining began in 1985, and during the process of mining, the boundary of the mining area has been continuously adjusted. For more than 30 years, a relatively mature "integration of mining, transportation, drainage and rehabilitation" was carried out [20]. In this study, four dumps in the Antaibao were selected as the study objects, namely, the South Dump, the West Dump, the West Expansion Dump, and the Inner Dump. Each dump has undergone a relatively long period of reclamation work, and currently forms a relatively stable ecosystem. The relevant information of each dump is shown in Table 1 [21–24].

**Table 1.** Basic information of the four dumps.

| Dump | Abandon Time | Formation Time | Reclamation Time | Main Vegetation Cover Types |
|---|---|---|---|---|
| South Dump | 1985 | 1989 | 1990 | tree |
| West Dump | 1985 | 1993 | 1994–1996 | shrub |
| West Expansion Dump | 1993 | 2005 | 2006–2008 | grass |
| Inner Dump | 1989 | The disposal work is expected to continue until 2059 | 1998–2001 | Mix of grass, shrub, and tree |

Due to dump occupation during mining, many environmental problems, such as terrain destruction, vegetation damage, and reduction in biodiversity, have continuously taken place, causing the dump to become a key site in land reclamation and ecological restoration [25]. In the Antaibao Open-pit Coal Mine, the integrated process of "mining-transport-dumping-reclamation" from west to east was adopted. Due to variable regional topography, dumps adopted different reclamation methods, leading to differences in the ecological restoration speed. Although the South Dump was first abandoned, the spontaneous combustion of coal gangue within the dump not only burned the vegetation, but also directly affected the soil moisture content, thus hindering the growth of surrounding vegetation. Later, the South Dump was reclaimed repeatedly, but the recovery speed was still slow. The West Dump was reclaimed in 1994 and 1996, with increasing vegetation coverage as the reclamation goal, but the recovery speed was also slow. In 2010, due to the industrial adjustment, the area occupied by the interference increased to 7.12% [23]. In the early formation of the West Expansion Dump, it was covered by natural forest, and later part of it was developed as an open-pit mining dump. Due to the short period of reclamation, there are still major disturbances and the ecosystem has not yet reached a stable state. The Inner Dump was an open pit in the early stage, and it was discharged after the pit is closed. Its recovery speed after reclamation was better than that of the other dumps.

*2.2. Data Collection*

In order to ensure the consistency of the vegetation spectrum information as much as possible, and in reference to existing research, August was selected for collecting images due to the excellent condition of the vegetation growth at this time [26]. Compared with other remote sensing satellites (such as ZY satellite and GF satellite), Landsat 8, Sentinel-2, and HJ satellites have earlier launch times and longer estimated service lives, and span a larger monitoring time. These are more suitable for long-term mining monitoring. The Landsat 8 and Sentinel-2 data used in the study were downloaded from the U.S. Geological Survey website (https://earthexplorer.usgs.gov/, accessed on 14 July 2021), and the HJ1A data were downloaded from the China Resources Satellite Application Center website (http://www.cresda.com/CN/, accessed on 22 July 2021). The relevant information of the obtained data is as follows (Table 2).

**Table 2.** Data list.

| Sensor | Spatial Resolution (m) | Time | Overall Cloud Coverage | Cloud Coverage within the Study Area |
|---|---|---|---|---|
| Landsat 8 OLI | 15, 30 | 15 August 2019 | 0.10% | 0% |
| Sentinel-2 MSI | 10, 20, 60 | 14 August 2019 | 3.94% | 0% |
| HJ1A CCD1 | 30 | 28 August 2019 | 0.00% | 0% |

The Spectral Response Function refers to the ratio of the received radiance to the incident radiance at each wavelength of the sensor. Due to the limitation of sensor hardware, the spectral response functions of different sensors are quite different, which has a very

direct impact on the reflectivity of vegetation [27]. In order to reflect the differences in sensors more clearly, the sensor-specific spectral response function is shown in Figure 2.

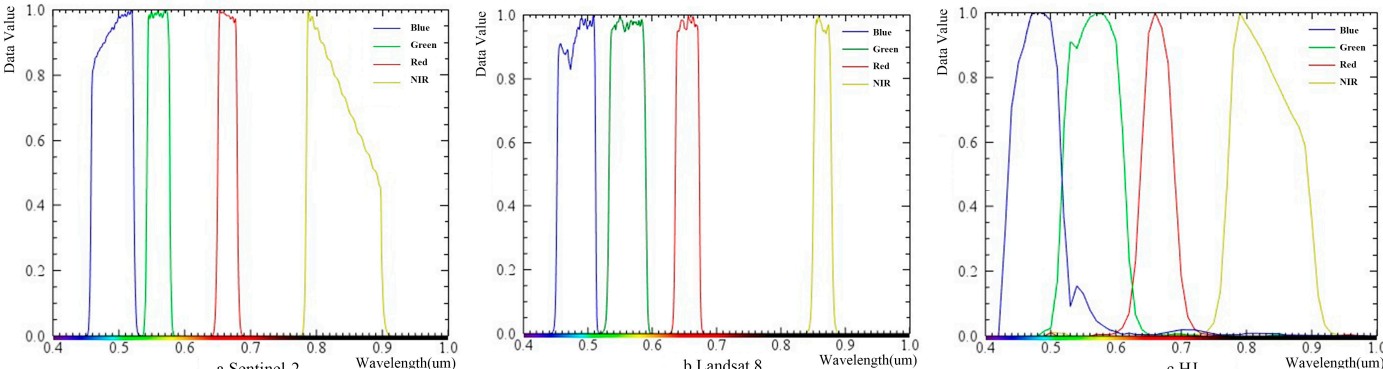

**Figure 2.** The spectral response function of the sensor ((**a**–**c**) are the spectral responses of Sentinel-2, Landsat 8, and HJ respectively).

Landsat satellites are jointly managed by USGS and NASA, and have the longest monitoring time. Since 1972, nine satellites have been launched, but Landsat 6 failed to launch, Landsat 7 was lost in 2012, and Landsat 8 was launched in 2013, equipped with OLI sensors. The 30 m visible light waveband, 15 m panchromatic waveband, and 30 m shortwave infrared waveband of Landsat enabled realization of long-term ground monitoring [28]. The Sentinel-2 optical satellite was launched in 2015, and managed by ESA (European Space Agency, Paris, France). It carried a high-resolution and multi-spectral image device to obtain 10 m visible light waveband, 20 m infrared waveband, 20 m red edge waveband, and 20 m shortwave infrared waveband information. The satellite is mainly used to monitor the growth of vegetation and land use cover. HJ (Environmental Disaster Satellite) was launched in 2008, and specially developed by China for detecting natural disasters. It is widely used in disaster monitoring and ecological assessment [29].

### 2.3. Method

In order to compare the correlation between sensor monitoring results under different types of vegetation cover (Table 1), NDVI monitored by HJ, Sentinel-2, and Landsat 8 were obtained separately. Based on the NDVI results, the correlations under different vegetation cover were obtained by unary linear regression analysis.

### 2.3.1. Preprocessing

In order to reduce the influence of sensor parameters, the image data from three satellites were preprocessed, including radiation calibration and atmospheric correction. After decompressing the Landsat 8 data, ENVI5.3 software was used for radiometric calibration to convert the DN value into a radiance value, and for FLAASH atmospheric correction to eliminate the influence of atmospheric factors and convert surface reflectance to land surface reflectance. Using Sen2Cor 02.09.00 software, Sentinel-2 data was subjected to radiation calibration and atmospheric correction. Sen2Cor software is a tool plug-in formatted for Sentinel-2 Level 2A products produced by ESA, which performs radiation calibration and atmospheric correction on L1C data. After atmospheric correction, SNAP software was used to convert the data format and resample images to 30 m spatial resolution to ensure the spatial resolution was consistent with that of other data. The original HJ data were subjected to radiometric calibration and band composition using patch tools, and the data after radiometric calibration were subjected to format conversion, geometric correction, and FLAASH atmospheric correction using ENVI5.3 software.

### 2.3.2. NDVI

The rapid development of remote sensing technology has enabled many vegetation indices to be obtained by formula calculations based on image band data, which has promoted the wide application of various vegetation indices in vegetation evaluation and monitoring [30]. The Normalized Difference Vegetation Index, reflecting characteristics of plant growth, vegetation coverage, and biomass, is widely used as an indicator to monitor vegetation growth. It has been used by many scholars both locally and globally for a long time to study the vegetation condition [31]. In this study, NDVI was selected as an indicator for vegetation monitoring to reflect the growth status of vegetation. Relevant studies have shown that this index can eliminate the influence of other factors (terrain, shadow, and atmosphere) to a certain extent.

As the most commonly used vegetation index, NDVI is defined as the ratio of the difference of the near-infrared and infrared bands to the sum of the near-infrared and infrared bands [32], and the threshold is [−1,1].

$$NDVI = \frac{(NIR - R)}{(NIR + R)}$$

where *NDVI* is the normalized difference vegetation index, *NIR* is the near-infrared band, and *R* is the visible red band. Generally speaking, when $-1 < NDVI < 0$, the land cover type is cloud, water, snow, etc., which are without vegetation cover; when $NDVI = 0$, the ground cover is rock or bare soil; when $0 < NDVI < 1$, the land is usually covered by vegetation. The larger the *NDVI* value, the better the vegetation condition. Based on the preprocessed remote sensing image, *NDVI* is calculated by the above formula.

### 2.3.3. Unary Linear Regression Analysis

Currently, regression analysis is a widely used predictive analysis method that can discover the association among variables through time series models, providing the correlation coefficient and linear regression equation [33,34]. Regression analysis originated from basic statistical concepts; based on the statistical results, related mathematical methods are used to obtain the basic assumptions, statistical inferences, and regression diagnosis. This is very useful for practical research, including dummy variables, interaction, auxiliary regression, polynomial regression, spline function regression, and step function regression.

Origin software has an effective data analysis function and is often used for analysis of statistical variables in scientific research and practical applications [35]. When calculating NDVI in this study, it was found that, due to the influence of design parameters and solar altitude angles of sensors, NDVI calculated by different sensors has certain differences. Exploring the differences between NDVI results from monitoring using different sensors is of great significance to the mutual substitution and supplementation of sensor data. In this study, based on the results of NDVI, 1000 random pixel points were established within dumps for regression analysis.

## 3. Results and Analysis

### 3.1. NDVI and Related Analysis in the South Dump

The South Dump was formed in 1985, and its reclamation started in 1990. It has experienced the longest reclamation time among the four dumps, and the physical and chemical attributes of the ecosystem are relatively stable. In the later reclamation work, due to improper operation, the coal gangue spontaneously combusted to cause damage to the reclaimed vegetation, which caused secondary damage to ecosystem. At present, the main type of vegetation coverage in the South Dump is trees. Figure 3 shows the NDVI results in the South Dump under the monitoring of three sensors.

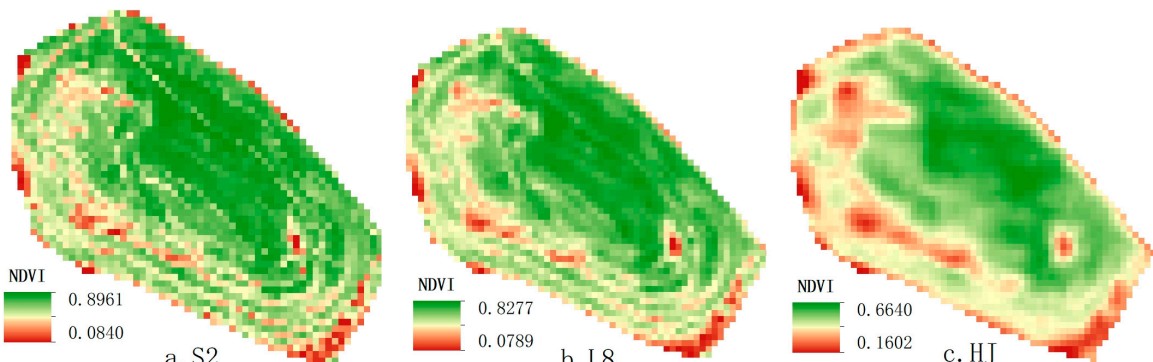

**Figure 3.** NDVI monitored by different satellite sensors in the South Dump ((**a**): NDVI caculated from Sentinel-2(S2); (**b**): NDVI caculated from Landsat 8(L8); (**c**) NDVI caculated from HJ).

According to Figure 3, the overall trend of NDVI in the South Dump under the monitoring of the three sensors is the same: showing a strip-like low value in the west, alternating yellow and red in the figure; a high value in the middle, shown as a large area in green in the figure; and a striped median value in the south, rendered as a red band in the figure. Judging from the actual situation, the strip-shaped low value in the west is the area of vegetation damage after reclamation due to the spontaneous combustion of coal gangue. The spontaneous combustion of coal gangue has always been a major problem in mining reclamation, and an effective solution is still being sought. From the perspective of the image rendering effect, the spatial effect of Sentinel-2 data in Figure 3a is the best because the original resolution of the Sentinel-2 data is the highest (10 m).

A total of 1000 random points were selected in the South Dump, and the correlation of random points' NDVI was analyzed under three types of sensors monitoring to obtain the conversion equations and correlation coefficients between every pair of sensors.

According to Figure 4, the fitting graphics of 1000 random points in the South Dump, itis observed that, in this area with trees as the main coverage type, Landsat 8 and Sentinel-2 have the best fitting effect and the highest correlation coefficient. From the NDVI value of random points, the number of abnormal points between Landsat 8 and Sentinel-2 is the least. Sentinel-2 and HJ sensors show a difference in NDVI results. When NDVI < 0.4, the fitting effect of the two sensors obviously shows that the value under HJ monitoring is higher than the value under Sentinel-2 monitoring. When NDVI > 0.5, the fitting effect of the two gradually moves closer to the fitting curve, indicating that the two types of sensors are more suitable for complementing each other in an area with high-density vegetation cover. The correlation coefficient between Landsat 8 and HJ is 0.8779, there are few abnormal points, and the fitting effect is ideal. The correlation coefficients and $R^2$ of Landsat 8, Sentinel-2, and HJ are summarized as follows (Table 3).

Taken together, the correlation coefficients between every pair of sensors in the South Dump were all above 0.8; notably, the correlation coefficient of Landsat 8 and Sentinel-2 sensors was 0.9266. This has certain reference significance for vegetation monitoring in areas covered by trees. However, the HJ and Sentinel-2 sensors are quite different, and were in the range of 0.4 < NDVI < 0.5, indicating that the two are not suitable for replacing and complementing each other in areas with medium-density vegetation coverage.

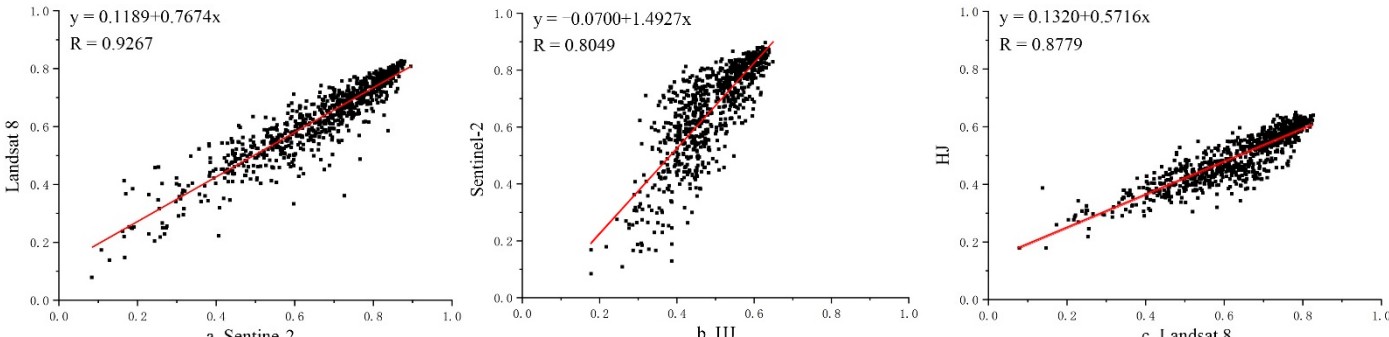

**Figure 4.** Correlation analysis of NDVI value based on random points (The figures include the conversion equation and correlation coefficient between every pair of sensors for NDVI. (**a**) shows the fitting result of NDVI between Landsat 8 and Sentinel-2; (**b**) shows the fitting result of NDVI between Sentinel-2 and HJ; (**c**) shows the fitting result of NDVI between HJ and Landsat-8).

**Table 3.** The NDVI fitting results in the South Dump under multi-source sensor monitoring.

| Satellite Sensor | $R^2$ | Correlation Coefficient |
| --- | --- | --- |
| Landsat 8 and Sentinel-2 | 0.8585 | 0.9267 |
| Sentinel-2 and HJ | 0.6476 | 0.8049 |
| HJ and Landsat 8 | 0.7704 | 0.8779 |

### 3.2. NDVI and Related Analysis in the West Dump

The West Dump began to be abandoned in 1985 and basically took shape in 1993. Its reclamation began in 1994 with increasing vegetation coverage as the main goal. The ecosystem is now relatively stable. The current vegetation coverage type is mainly shrub. Figure 5 shows the NDVI results in the West Dump under the monitoring of the three sensors.

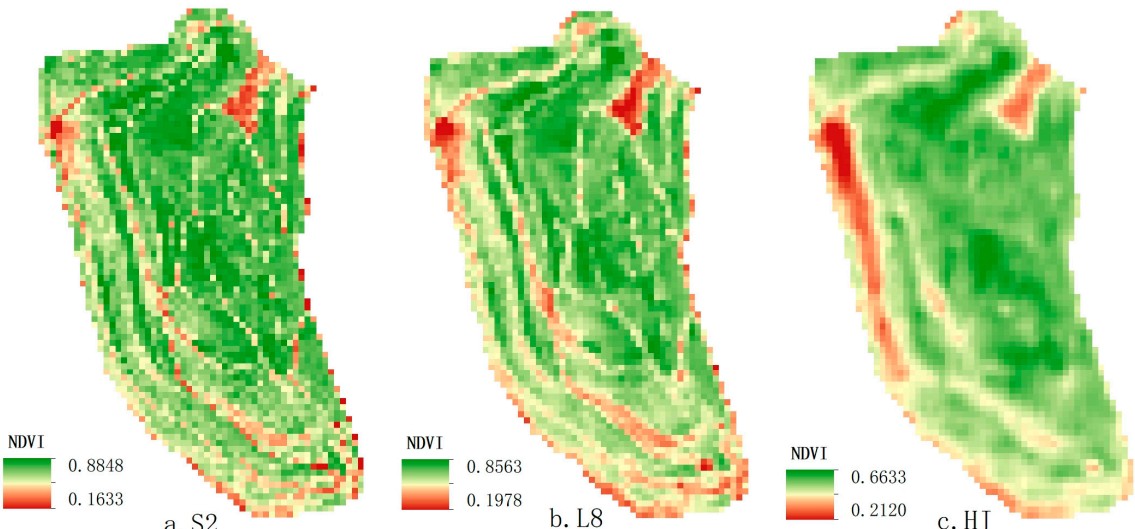

**Figure 5.** NDVI monitored by different satellite sensors in the West Dump ((**a**): NDVI caculated from Sentinel-2(S2); (**b**): NDVI caculated from Landsat 8(L8); (**c**) NDVI caculated from HJ).

According to Figure 5, from an overall point of view, the overall trend of NDVI under three sensor monitoring is consistent, showing a strip-like low value in the west, a high-value distribution trend in most of the central and eastern regions, and a small regional low-value area in the northeast. Due to the high accuracy of the Sentinel-2 original image

data, the linear distribution within the dump in Figure 5a can be observed more clearly. It is estimated that the linear distribution is the transportation road within the West Dump.

A total of 1000 random points were selected in the West Dump, and the correlation of the NDVI of the random points was analyzed under the monitoring of the three sensors to obtain the conversion equations and correlation coefficients between every pair of sensors.

According to Figure 6, the fitting graphics of 1000 random points in the West Dump, it is observed that in this area with shrubs as the main coverage type, NDVI values under Landsat 8 and Sentinel-2 monitoring were mostly higher than 0.4, showing that the West Dump had a high-density vegetation coverage. In addition, the coefficient of the equation between Landsat 8 and Sentinel-2 was 0.6610, which is less than 1, indicating that the NDVI under Sentinel-2 monitoring is generally larger than that of Landsat 8 monitoring. HJ and Sentinel-2 sensors do not show an obvious fitting effect when NDVI < 0.4. However, when NDVI increases above 0.4, the fitting effect gradually moves closer to the fitting curve. The fitting effect of Landsat 8 and HJ is also improved when NDVI > 0.5, but there are some outliers. The correlation coefficients and $R^2$ of Landsat 8, Sentinel-2 and HJ are summarized in Table 4.

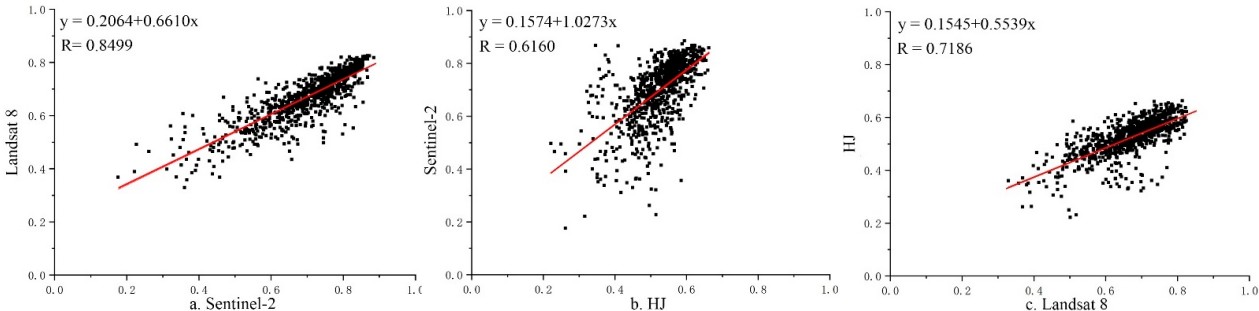

**Figure 6.** Correlation analysis of NDVI value based on random points (The figures include the conversion equation and correlation coefficient between every pair of sensors for NDVI. (**a**) shows the fitting result of NDVI between Landsat 8 and Sentinel-2; (**b**) shows the fitting result of NDVI between Sentinel-2 and HJ; (**c**) shows the fitting result of NDVI between HJ and Sentinel-8).

**Table 4.** The NDVI fitting results of the West Dump under multi-source sensor monitoring.

| Satellite Sensor | $R^2$ | Correlation Coefficient |
| --- | --- | --- |
| Landsat 8 and Sentinel-2 | 0.7220 | 0.8499 |
| Sentinel-2 and HJ | 0.3788 | 0.6160 |
| HJ and Landsat 8 | 0.5158 | 0.7186 |

Taken together, the correlation coefficients of NDVI under the monitoring of the three sensors in the area covered by shrubs shows a large difference: the correlation coefficient between Landsat 8 and Sentinel-2 sensors was also the highest, as in the case of the South Dump, and the fitting effect was still the best. Relatively speaking, the correlation coefficient between HJ and Sentinel-2 sensors was lowest in West Dump. Although the correlation coefficient between Landsat 8 and HJ sensor was at a medium level, there were still some abnormal points.

### 3.3. NDVI and Related Analysis in the West Expansion Dump

The West Expansion Dump began to be abandoned in 1985 and basically took shape in 1993. The ecological environment is now relatively stable, and the current vegetation cover type is mainly grass. Figure 7 shows the NDVI results in the West Expansion Dump under the monitoring of the three sensors.

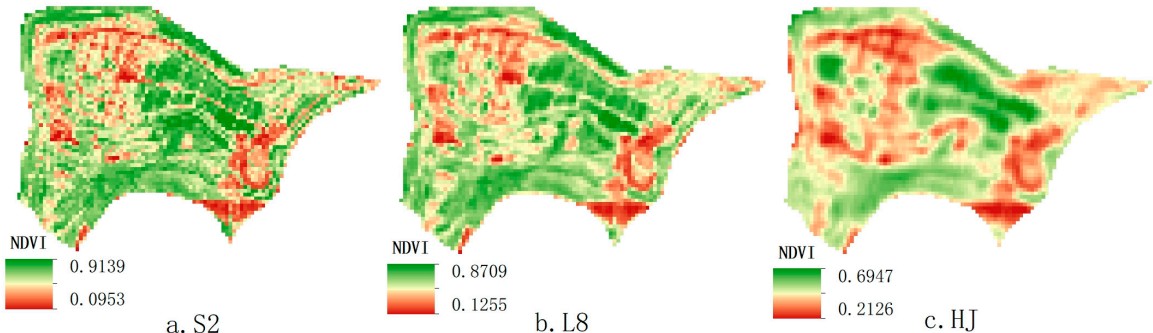

**Figure 7.** NDVI monitored by different satellite sensors in the West Expansion Dump ((**a**): NDVI caculated from Sentinel-2(S2); (**b**): NDVI caculated from Landsat 8(L8); (**c**): NDVI caculated from HJ).

According to Figure 7, the overall trend of NDVI under three sensor monitoring is similar, but the NDVI results under HJ monitoring are not ideal for clearly distinguishing the high and low values. Sentinel-2, having the highest original resolution, has the highest sensitivity to NDVI and the best rendering effect among the three sensors. The monitoring results of Sentinel-2 can enable the high and low values of NDVI to be distinguished more clearly. Additionally, the distribution of roads within the dump can be clearly observed under Sentinel-2 monitoring.

A total of 1000 random points were selected in the West Expansion Dump, and the correlation of NDVI of the random points was analyzed under the monitoring of the three sensors to obtain the conversion equations and correlation coefficients between every pair of sensors.

According to Figure 8, the fitting graphics of 1000 random points in the West Expansion Dump, in this area with grass as the main coverage type, NDVI under Landsat 8 and Sentinel-2 monitoring was closely arranged near the fitting curve, which reflects that the two sensors have a high correlation when monitoring grass. The monitoring results of the HJ and Sentinel-2 sensors clearly show that NDVI under Sentinel-2 monitoring is higher than that of HJ monitoring because the conversion equation coefficient is more than 1. The correlation coefficient between the two is only 0.6495, and the linear fitting cannot fit most points very well. The fitting curve of Landsat 8 and HJ also shows that NDVI values under HJ monitoring are generally lower. The correlation coefficients and $R^2$ of Landsat 8, Sentinel-2, and HJ are summarized Table 5.

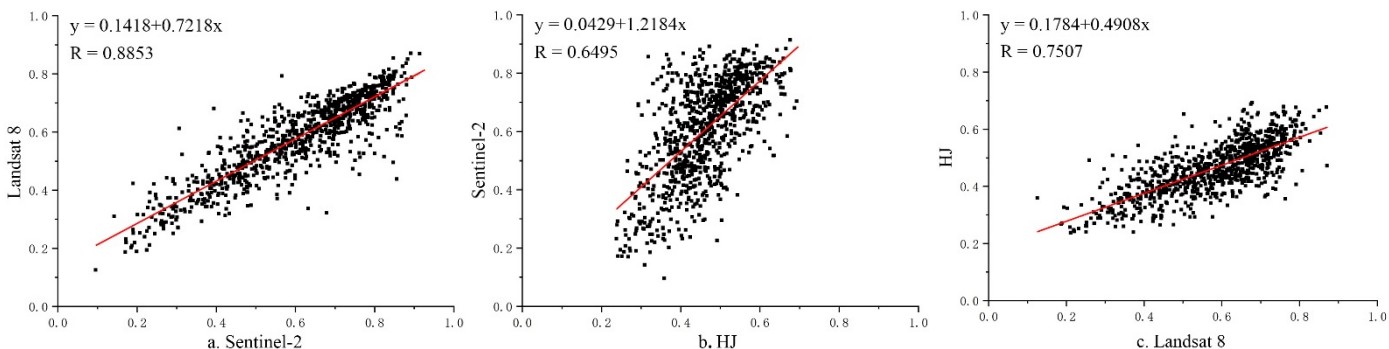

**Figure 8.** Correlation analysis of NDVI value based on random points (The figures include the conversion equation and correlation coefficient between every pair of sensors for NDVI. (**a**) shows the fitting result of NDVI between Landsat 8 and Sentinel-2; (**b**) shows the fitting result of NDVI between Sentinel-2 and HJ; (**c**) shows the fitting result of NDVI between HJ and Landsat 8).

**Table 5.** The NDVI fitting results of the West Expansion Dump under multi-source sensor monitoring.

| Satellite Sensor | $R^2$ | Correlation Coefficient |
|---|---|---|
| Landsat 8 and Sentinel-2 | 0.7836 | 0.8853 |
| Sentinel-2 and HJ | 0.4213 | 0.6495 |
| HJ and Landsat 8 | 0.5631 | 0.7507 |

Taken together, the correlation coefficients when multi-sensor monitoring is applied to grass in the West Expansion Dump are close to the results when multi-sensor monitoring is applied to shrubs in the West Dump; that is, correlation coefficients between Landsat 8 and Sentinel-2 > correlation coefficients between Landsat 8 and HJ > correlation coefficients between HJ and Sentinel-2.

*3.4. NDVI and Related Analysis in the Inner Dump*

The Inner Dump began to be discarded in 1989, and it is expected that the discarding work will continue until 2059. It is in a state of reclamation while being currently abandoned. The vegetation cover type is mainly a mix of grass, shrubs, and trees. Figure 9 shows the NDVI results in the Inner Dump under the monitoring of the three sensors.

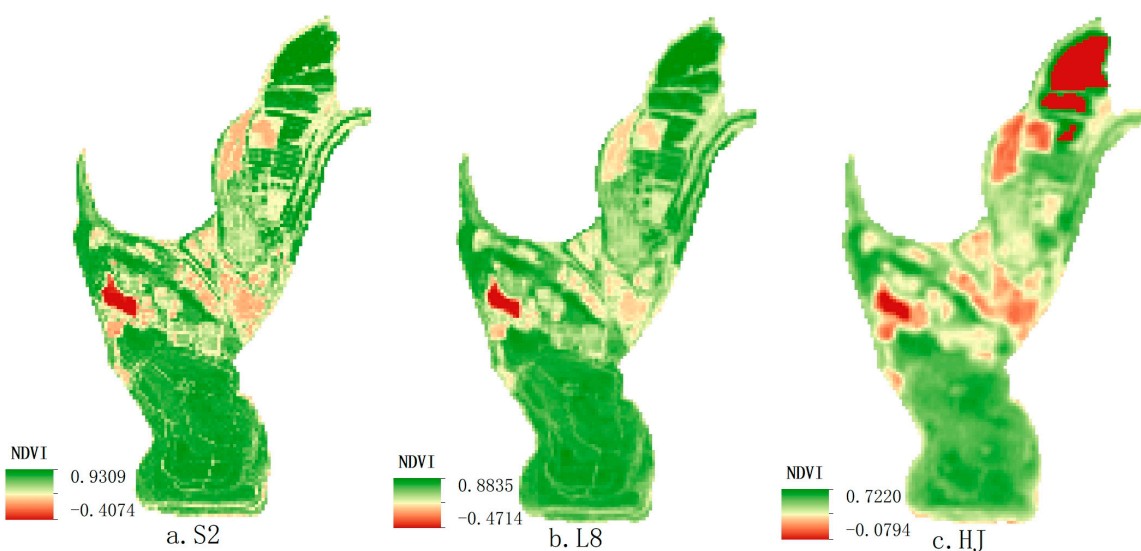

**Figure 9.** NDVI monitored by different satellite sensors in the Inner Dump ((**a**): NDVI caculated from Sentinel-2(S2); (**b**): NDVI caculated from Landsat 8(L8); (**c**) NDVI caculated from HJ).

According to Figure 9, the overall trend of NDVI results under the monitoring of the three sensors is consistent in most areas. However, the results under HJ monitoring have obvious abnormal values in the northeast region. This t should be an area with high NDVI values, but it has a low value distribution. After inspection, it was found that the original image pixels were missing. From the perspective of NDVI distribution, Sentinel-2 data and Landsat 8 data are very similar, they have a better effect on distinguishing high values and low values, and the vegetation distribution inside the dump can be clearly observed. Furthermore, it should be pointed out that the detail of Sentinel-2 is the clearest of the three sensors.

A total of 1000 random points were selected in the Inner Dump, and the correlation of random points' NDVI was analyzed under the monitoring of the three sensors to obtain the conversion equations and correlation coefficients between every pair of sensors.

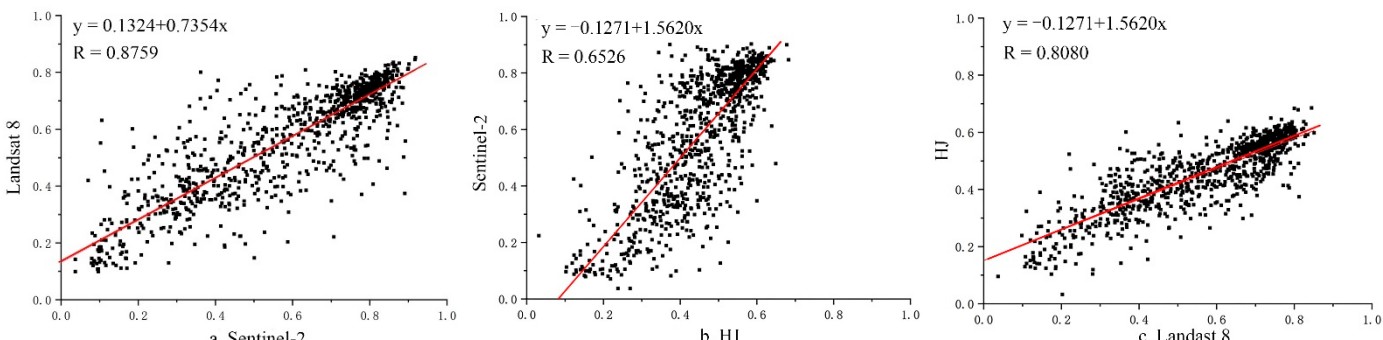

**Figure 10.** Correlation analysis of NDVI value based on random points (The figures include the conversion equation and correlation coefficient between every pair of sensors for NDVI. (**a**) shows the fitting result of NDVI between Landsat 8 and Sentinel-2; (**b**) shows the fitting result of NDVI between Sentinel-2 and HJ; (**c**) shows the fitting result of NDVI between HJ and Landsat 8).

According to Figure 10, the fitting graphics of 1000 random points in the Inner Dump, it is observed that, in this area where the main vegetation coverage type is a mix of grass, shrubs, and trees, there are generally more irregularly distributed points when performing correlation analysis between any pair of sensors. However, a large number of discrete points does not have a large effect on the correlation coefficient. The correlation coefficient between Landsat 8 and Sentinel-2 is close to 0.9, the correlation coefficient between Landsat 8 and HJ is higher than 0.8, and the correlation coefficient between Sentinel-2 and HJ exceeds 0.6. The correlation coefficients and $R^2$ of Landsat 8, Sentinel-2, and HJ are summarized in Table 6.

**Table 6.** The NDVI fitting results of the Inner Dump under multi-source sensor monitoring.

| Satellite Sensor | $R^2$ | Correlation Coefficient |
| --- | --- | --- |
| Landsat 8 and Sentinel-2 | 0.7670 | 0.8759 |
| Sentinel-2 and HJ | 0.6526 | 0.6526 |
| HJ and Landsat 8 | 0.7519 | 0.8080 |

On the whole, the fitting results of NDVI under three sensor monitoring were more controversial in the Inner Dump. Although there were many abnormal points, the correlation coefficient was not the lowest, indicating that the overall fitting effect was relatively good. Objectively speaking, the reclamation time of the Inner Dump occurred later than that of the West Dump, but the fitting effect is better than that of the West Dump, indicating that fitting effect and reclamation time have no direct connection.

### 3.5. Comprehensive Analysis of Multi-Source Satellite Monitoring Results

According to the random point information, statistical characteristic values, including the maximum, minimum, and average values of NDVI monitored by multi-source satellites within each dump, were derived.

According to Table 7, under the same vegetation type coverage, the monitoring results under different sensors were different; under different types of vegetation coverage, the correlations of sensor monitoring results were also different.

From the average value of the correlation coefficients, the correlation of satellite monitoring results is affected by the vegetation cover type, showing that correlation from strong to weak was: tree > mix of grass, shrub, and tree > grass > shrub.

From the point of view of reclamation time, the South Dump began to be abandoned first, and the long-term restoration has formed a relatively stable ecosystem. The reclamation time of the Inner Dump is similar to that of the West Dump, but the correlation coefficients of the satellite monitoring results are higher than those of the West Dump. The West

Dump has been restored to a certain extent after reclamation, but has not yet reached a stable state due to disturbance caused by man-made or other factors. Although the recovery time after reclamation within the West Expansion Dump is the shortest, the correlation coefficients of the satellite monitoring results are not the lowest, and are close to the results of the Inner Dump.

**Table 7.** NDVI statistical characteristic values based on different vegetation cover types.

| Dump | Main Vegetation Cover Type | Sensor | Min | Max | Mean | Mean Value of Correlation Coefficient |
|---|---|---|---|---|---|---|
| South Dump | tree | LNDVI | 0.0790 | 0.8278 | 0.6280 | 0.8698 |
| | | SNDVI | 0.0840 | 0.8961 | 0.6607 | |
| | | HNDVI | 0.1603 | 0.6641 | 0.4892 | |
| West Dump | shrub | LNDVI | 0.1978 | 0.8563 | 0.6645 | 0.7282 |
| | | SNDVI | 0.1633 | 0.8848 | 0.6945 | |
| | | HNDVI | 0.2121 | 0.6634 | 0.5234 | |
| West Expansion Dump | grass | LNDVI | 0.1255 | 0.8709 | 0.5754 | 0.7619 |
| | | SNDVI | 0.0953 | 0.9139 | 0.5988 | |
| | | HNDVI | 0.2127 | 0.6948 | 0.4632 | |
| Inner Dump | Mixed coverage | LNDVI | −0.4715 | 0.8836 | 0.5619 | 0.7788 |
| | | SNDVI | −0.4075 | 0.9309 | 0.5865 | |
| | | HNDVI | −0.0795 | 0.7221 | 0.4632 | |

From the mean of NDVI, the monitoring values of Landsat 8 and Sentinel-2 sensors are similar for every vegetation cover type, while the values under HJ monitoring are relatively lower. Additionally, from the point of view of NDVI value, NDVI monitored by the HJ is generally in the middle range. That is, compared with Landsat 8 and Sentinel-2 satellites sensors, NDVI values under HJ monitoring show a phenomenon in which the low values are higher and the high values are lower.

## 4. Discussion

### 4.1. Analysis of the Influence of Satellite Surface Reflectance

Differences in atmospheric conditions, observation angles, and surface morphology during imaging by different satellite sensors are likely to cause systematic errors in apparent reflectance, thus further affecting the results of NDVI in the later calculations [36]. In Figure 2, through the spectral curves of the three satellite sensors, it can be seen that the spectral curve characteristics of Sentinel-2 and Landsat 8 have high consistency in the blue and red bands. Compared with Landsat 8, the Sentinel-2 satellite sensor has a wider wavelength in the near-infrared band and a narrower wavelength in the green band. However, the spectral curve characteristics of the HJ satellite sensors are very different, showing that the wavelengths of the visible light band and the near-infrared band are relatively wide. The spectral response function of HJ causes the phenomenon of "Low value is high and high value is low" in vegetation monitoring, which indicates that the HJ sensor is weaker than other satellite sensors in acquiring vegetation information. Differences in spectral response functions result in differences in the reflection signals of objects received by the sensor in the red and near-infrared bands.

### 4.2. Analysis for Influence of Spatial Resolution

The original spatial resolution of Sentinel-2 is 10 m, and the original spatial resolution of Landsat 8 and HJ is 30 m. Although resampling was undertaken, the results in Figure 3 indicate that different data have different abilities to distinguish the details of ground objects. The higher the spatial resolution of the image, the easier it is to capture the spectral information of small objects.

*4.3. Analysis for Influence of Transit Time*

Although the satellite image data were obtained within the same month, changes in atmospheric conditions also have a certain impact on the cross-comparison results. The solar zenith angle will change with the satellite transit time, which will also affect the inversion results of apparent reflectance [37].

**5. Conclusions**

Through the interactive comparison of Landsat 8, Sentinel-2, and HJ sensors, it was found that the correlations of different sensors in monitoring vegetation make a significant difference to the results. The conclusions are as follows:

(1) Landsat 8 and Sentinel-2 data have a high correlation for monitoring vegetation, but the correlation between the above two sensors and the HJ sensor is relatively low, and the difference between Sentinel-2 and HJ is more obvious. The mean correlation coefficient between Landsat 8 and Sentinel-2 is 0.8845, the mean correlation coefficient between Sentinel-2 and HJ is 0.6808, and the mean correlation coefficient between HJ and Landsat 8 is 0.7888.

(2) The results of correlation analysis show that the correlation of sensor monitoring results changes with the change in vegetation coverage. From strong to weak, it is ordered as: tree in the South Dump (R = 0.8698) > combined grass, shrub, and tree in the Inner Dump (R = 0.7788) > grass in the West Expansion Dump (R = 0.7619) > shrub in the West Dump (R = 0.7282).

(3) The phenomenon of "Low value is high and high value is low" in the NDVI value under HJ monitoring may be caused by the weaker signal strength of the HJ sensor in the visible light band.

It should be pointed out that the conversion equations of multi-source remote sensing data obtained in this study are only applicable to the specific vegetation cover type in the Loess Plateau, and the atmospheric conditions are good. In addition, there are some problems, such as loss of pixel information and pixel distortion of the original environmental satellite data, which affect the correlation analysis. Additionally, it should be pointed out that NDVI has some limitations. Although NDVI can reflect vegetation growth well, the monitoring accuracy is directly affected by the development of remote sensing technology. In addition, cloudy and foggy weather in some areas has an impact on the quality of remote sensing images, which directly leads to large fluctuations in NDVI data, resulting in errors in vegetation identification. Furthermore, related research shows that NDVI has lower sensitivity in areas with high vegetation density [38].

**Author Contributions:** Conceptualization, J.H. (Jiameng Hu) and B.Y.; methodology, J.H. (Jiameng Hu) and B.Y.; formal analysis, J.H. (Jiameng Hu); investigation, J.H. (Jiameng Hu) and B.Y.; resources, J.H. (Jiameng Hu) and B.Y.; data curation, J.H. (Jiameng Hu); writing—original draft preparation, J.H. (Jiameng Hu); writing—review and editing, Z.B., B.Y. and J.H. (Jiawei Hui); su-pervision, Z.B. and B.Y.; project administration, Z.B.; funding acquisition, Z.B. All authors have read and agreed to the published version of the manuscript.

**Funding:** This research was funded by budget items of the Department of Territorial Space Ecological Restoration of the Ministry of Natural Resources, People's Republic of China "Study on Adaptive Management of Ecological Protection and Restoration of mountains, rivers, forests, fields, lakes, grasses" (HBA056).

**Institutional Review Board Statement:** Not applicable.

**Informed Consent Statement:** Not applicable.

**Data Availability Statement:** The remote sensing data used in this article are public and can be found at https://earthexplorer.usgs.gov/ (accessed on 14 July 2021) and http://www.cresda.com/CN/ (accessed on 22 July 2021).

**Conflicts of Interest:** The authors declare no conflict of interest.

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
