# Peer review of "Comparison of the Vegetation Index of Reclamation Mining Areas Calculated by Multi-Source Remote Sensing Data"

_land, doi:10.3390/land11030325_

Round 1

Reviewer 1 Report

Dear Authors,

Happy New Year! Hope the new year brings you more happiness, joy and good health!

I had the pleasure of reviewing your manuscript and have some feedback for you.

The manuscript presents a comparative analysis of the NDVI derived by multi-sensoral data in mining area. Actually, there has been a multitude of studies focusing on comparison of the vegetation indices derived from different satellite data for monitoring vegetation, especially from Sentinel-2 MSI and Landsat-8 OLI (see Zhang et al., 2018. Characterization of Sentinel-2A and Landsat-8 top of atmosphere, surface, and nadir BRDF adjusted reflectance and NDVI differences). However, this paper didn’t present more novel research.

The introduction section needs to be written to fully illustrate the significance of remote monitoring the vegetation of dump in mining area with multi-sensor. The materials and methods section needs a figure to depict sensor-specific spectral response functions. The results and analysis section needs to firstly discuss the differences in satellite surface reflectance, particularly which were used for NDVI. Furthermore, authors should enlarge the discussion in such a way as to reveal several unique advantage for monitoring and evaluating restoration vegetation status in mining area using multi-sensoral remotely sensed data.

Reviewer 2 Report

First and foremost, I would like to congratulate the Authors for writing a concise, yet straight to the point manuscript. Short and straighforward articles are rarely seen and for that the Authors have done a very good job. However, there are some elements that need to addressed:

1.Abstract
a. The use of English should be improved and some numerical results should be added. The objectives and novelties should also appear better.

2. Introduction
a. Please state better the study's novelties, scientific gaps it aims to bridge and its contribution to the field. 

3. Research area
a. What do you mean by arbor? If it refers to trees better use the term trees then

4. Data collection
a. There is no need for the historical narrative on LANDSAT
b. SENTINEL 2 is still relatively young. LANDSAT has the upper hand on all satellites in terms of historical coverage please mention this and show LANDSAT's superiority in terms of temporal coverage.

5. Section 2.3.2
a. The NDVI formula has brackets please add them.

6. Results
a. There should be an additional section that discusses the limitations of NDVI. If the Authors did not consider these the study's results can become questionable
b. The resolution of the maps is quite poor it needs to be enhanced. While this might be due to the relatively small size of the study area with respect to the satellites' spatial resolution, better representations are needed.
c. How do the Authors propose a validation to NDVI findings while it is widely known that NDVI validation methods are still very limited even to non-exsitent?
d. Using August as the time frame only means that a more or less snapshot effect is obtained. A single snapshot, whereas the whole vegetation growth cycle should be covered. How did the Authors account for atmospheric effects on NDVI ? (abundance or absence of rain)? How did they account for its saturation above densely vegetated areas? How can the Authors assume recoveries just using NDVI? I am afraid that are many assumptions that need to be validated or proved by literature. 
If I understand correctly only 2019 images were used? Then again how are the Authors comparing satellites with different resolutions? 

Reviewer 3 Report

This is a very interesting paper that reviews the use of different satellite sensors for assessing mineland dump sites and the scientific application of remote sensing for quantifying vegetation condition and change in relation to mining is insightful. The authors use NDVI to assess vegetation greenness, as a correlate to vegetation cover. I think the application of remote sensing to assess reclamation success is a valuable and important consideration. This paper uses what apprear to be appropriate techniques for quantifying NDVI and produce results that show direct correlation between sensors and greenness values at each location. To me, these are all good things. Areas that need attention are as follows: 

  1. the most important issue is that much of the paper requires improvement in sentence structure and word choice. Often the meaning of a sentence or intent of that information is confused or lost in the inaccurate use of the English language. This will need attention to improve clarity and the paper generally
  2. The difference between the information presented in the results and discussion section are difficult to differentiate. The discussion lacks references to other literature or research to build onto their own work, or to support or refute other's work. 
  3. It would be very helpful to have a more detailed explanation of the vegetation composition at each site. It uses general terms of grass, shrubs, etc.., but what those differences between vegetation composition and structure is not clear. How do the ecosystems between all four sites compare? Are the species of plants similar or highly different? Are there invasive species issues that could influence the NDVI values? Are the soils and climate comparable? More information is needed.
  4. The figures lack complete caption descriptions. These captions should be expanded and improved for clarity and enough information provided to be stand alone figures.  

Following are a number of specific edits and comments that should be addressed prior to being considered for publication:

Title: Consider the title “Comparison of Vegetation Indices within Reclaimed Mining Areas Calculated using Multi-source Remote Sensing Data”

Page 1, Line 11: consider rewriting this sentence to read “After reclaiming vegetation within a mining area, secondary damage may occur at any time, particularly within areas that have been mined for decades or hundreds of years.” Also, its not clear what you mean by “secondary damage”. What is that?

Page 1, Line 12: consider rewriting this sentence as something like “It is necessary to accurately monitor vegetation growth to effectively assess revegetation success and influence.” Otherwise the wording is a little confusing and not clearly written. In particular, I am not sure what you mean by “completing the monitoring task” and “reflecting the effect”.

Page 1, Line 17: rather than saying “In this text, …” I would recommend you state “In this study, …”

Page 1, Line 20: I think the word satellite should be made plural to read “satellites”. Also, it is not totally clear what you mean by “high relevance”. 

Page 1, Line 21-22:  would recommend rewording this to “2) the correlation between vegetation and the sensors used varied by vegetation type. The correlation ranged from strong to weak: …”

Page 1: for this abstract, it would be helpful to end this paragraph is a description of what these results mean (how are they applied) or what is their significance.

Page 1, Introduction:

The introduction section provides a relatively thorough assessment of background and associated literature. The objectives statement at the end of the introduction provides a good description of what the justification for this study and the use of NDVI to quantify vegetation condition. Having said this, my primary concern for this paper is the need for greater clarity in the use of the English language. Most of the writing is understandable, however, there is occasionally some confusion in what is being said.  More common is that this limitation impacts sentence structure, clarity in meaning, and not being sure the intent of what is being said. I have provided a number of recommendations for improving the writing below, but this level of review should be provided throughout the paper. I understand most of what is being said, but there is some confusion on the intent of sentences because of a lack of clarity in the writing.

Page 1, Line 29: this first sentence is not clearly worded. I would recommend something like “Coal has played a vital role in the China energy production demands. Coal is critical for providing energy for sustaining life and in the production of good and services.”

Page 1, Line 36: the statement “…long-term land reclamation in the long-term mining process…” is wordy. I would rewrite this sentence to be more concise in general.

Page 1, Line 37: this sentence could be rewritten to state something like “Using effective monitoring techniques within mining areas, vegetation condition can be captured enhancing restoration planning and implementation.”

Page 1, Line 40: I would avoid the word “huge”. This term is relative and not typically used in scientific writing. A more common word is “significant” or “tremendous”.

These types of wording issues are common in this paper. It would be helpful to edit the text to improve sentence structure and enhance clarity/understanding.

Page 2, Line 82-83: It would be advised to include actual average values for precipitation (snow/rain) and temperature (range or max/min).

Page 2, Line 85: please be more descriptive of the word “arbor”. If this is represented by certain species of  plants, provide their common/scientific names. On line 86, state if the grass is annual or perennial. If you know if it is native or introduced or invasive, this information would be helpful. You state these plant categories, but it would be meaningful to include some of the species that make up these groups. For example, what are the dominant shrubs that you are measuring? This is important when considering remote sensing of vegetation since shrubs can reflect light differently.

Page 3, Line 97-100: are these four sites ecologically comparable? Do they have similar soils, elevation, precipitation levels, etc? It would be helpful to know how they compare.

Page 3, Line 111: are you referring to high air temperature when you say that the temperature affected plant growth? Is the temperature affecting it, or is the soil moisture also impacted by increased temperatures that could be impacting vegetation growth?

Page 4, Line 124: be specific with why you chose August and what you mean by “the vegetation is better”. How is it better (what variables did you measure for determine this)?

Page 5: The NDVI segment of the methods is clearly written and effectively described. Nice summary of this application.

Page 5, Line 204-207: This paragraph sounds better suited as methods than results.

Page 6, Line 218: This figure caption needs to effectively describe the information presented in the figure. Its not self-standing as it is currently written. This needs to be improved for providing greater clarity and location information.

Page 6, Line 219-222: this description of figure 2 is unclear. It would help to be more specific what you mean by strip-like low value or striped median. I am not saying that this description is not useful, but it needs greater clarity what you mean and what these terms imply.

Page 6, Line 223: same issue with figure caption 3. This is interesting data that needs to be included, but a more complete description in the caption of what these data represent, any definition of acronyms (HJ), and location are needed here. These same issues apply to all figures in the paper, esp. since the structure is redundant by design (discussing each site individually). If a more clear description is provide the first site results (south dump), then the other sites could be more generalized.

Page 11, Line 382-384: This is another example of how poor sentence structure and word choice makes this information difficult to understand. I would recommend rewriting this sentence to something like “Based on differences in vegetation cover between each of the different dumps, a linear fitting analysis was performed on NDVI values comparing different monitoring sensors. It was determined that the correction between these monitoring results varied by differences in vegetation type.” I am not sure this is exactly what you are trying to say, but improving the wording will make the interpretation of the results more straight forward and comprehendible.

Page 11, Line 385: compared to arbor (highest) and mixed veg (next best), how do the grass/shrub results compare to these others?

Page 12, Line 399-423: these are meaningful data, but they read a lot to me like results. I think it would be important to further interpret these results and relate them to other papers/citations. There are only a couple papers cited in the entire discussion section, and this is typically an important aspect of an effective discussion – comparing your work with others (both positive comparisons and also how they differ or tell the story in a different way).

Round 2

Reviewer 2 Report

The Authors have provided many changes and have addressed well the comments. The only minor element is the use of English. I propose a revision by a Native English speaker or though MDPI's language service. Other than that, the mansucript is suitable. However, it should present a better use of English for international readership

Author Response

Dear reviewer,

First of all, thank you very much for your patient guidance!

As you said, some English expressions of the article need to be improved. I thoroughly checked the sentence structure and grammar in the article and found many inappropriate expressions in the article. I was ashamed of it and have made a careful revision.

Yours sincerely!

Reviewer 3 Report

I appreciate the improvements that you have made on this paper. The purpose of the research is clear and the methods are appropriate for the type of analysis you did for this study. You have addressed the comments I had made and the paper and its is definitely improved. 

While the wording and sentence structure from the first draft is greatly improved, there is still a need to correct the use of the English language in this paper. I understand that this must be frustrating since English is not your first language (as mentioned in the responses to my initial comments). I have included here some edits in a word document that I think could help improve the paper by creating greater clarity and improving grammar and word structure. 

In response to providing a rapid turn-around, its only a partial review. I hope this helps and gives you a gauge for the type of edits that I think are needed to get the manuscript prepared.  

Author Response

Dear reviewer,

Thank you very much for your valuable comments and careful review.

Your careful review has helped me to discover many insufficient in English expression. So I have thoroughly checked and revised the sentence structure and grammar of the article. And in the revision, I improved my English expression ability.

Thanks again for your patient guidance.

Yours Sincerely!
